# Future Technologies for Train Communication: The Role of LEO HTS Satellites in the Adaptable Communication System

**DOI:** 10.3390/s23010068

**Published:** 2022-12-21

**Authors:** Alessandro Vizzarri, Franco Mazzenga, Romeo Giuliano

**Affiliations:** 1Department of Engineering Science, Guglielmo Marconi University, 00198 Rome, Italy; 2Radiolabs Consortium, Department of Enterprise Engineering, University of Roma Tor Vergata, 00133 Rome, Italy

**Keywords:** low-earth orbit satellite, 5G, railway, ACS, FRMCS

## Abstract

The railway sector has been characterized by important innovations regarding digital technologies for train-to-ground communications. The actual GSM-R system is considered an obsolescent technology expected to be dismissed by 2030. The future communication systems in the rail sectors, such as Adaptable Communication Systems (ACS) and Future Railway Mobile Communication Systems (FRMCS), can manage different bearers as 4G/5G terrestrial technologies and satellites. In this environment, the new High Throughput Satellite (HTS) Low-Earth Orbit (LEO) constellations promise very interesting performances from data rate and coverage points of view. The paper analyzes the LEO constellations of Starlink and OneWeb using public data. The Rome–Florence railway line is considered for simulations. The results evidence the LEO satellite can provide interesting performance in terms of visibility, service connectivity, and traffic capacities (up to 1 Gbps). This feature enables the LEO to fully manage a high amount of data, especially in the railway scenarios of the next years when video data applications will be more present.

## 1. Introduction

The European Green Deal defines important objectives in the transport sector, in particular, climate neutrality by 2050. Involving advanced telecommunication technologies in the rail sector allows not only increasing safety and security of the transport sector but also a reduction in the emissions of CO_2_. The telecommunication infrastructure is crucial for future train management systems such as European Rail Traffic Management System/European Train Control System (ERTMS/ETCS) in Europe.

The ERTMS/ETCS are a crucial component of the Adaptable Communication System (ACS) and Future Railway Mobile Communication System (FRMCS), able to use different communication technologies.

The increase in demand for communication and signaling applications requires broadband communication systems allowing accurate train positioning, automatic train operation (ATO), and efficient predictive maintenance of rail infrastructure. The 4G/5G terrestrial networks can be affected by: (i) the limitation of the available radio spectrum and (ii) the necessity to identify efficient communication technologies able to overcome the obsolescence of the actual communication standard (GSM-R). The end of life for GSM-R is expected to be around 2030.

Satellite technologies are broadband communication systems able to integrate terrestrial networks and even to substitute them when terrestrial is not available.

The Geostationary Earth Orbit (GEO) and Medium-Earth Orbit (MEO) satellites are already used in the rail sector also for communication applications from/to the train. However, the considerable values of end-to-end delay limit the usage of GEO and MEO in the rail environment [1,2].

Both terrestrial communication systems (4G/5G) and GEO/MEO satellite constellations are called Traditional Bearers (TB).

Due to the previous factors, it is necessary that additional bearers, called Alternative Bearers (AB), should be identified. Among these ABs, the LEO new constellations arose a crescent interest thanks to the improvement of performance concerning GEO/MEO satellites.

The paper is organized as follows. Section 2 presents the state of the art of usage of LEO constellations in the rail sector. Section 3 describes the railway scenarios and applications present in the railway context. Section 4 analyzes the architectures of ACS and FRMCS systems. Section 5 reports the LEO satellites’ role within the FRMCS/ACS infrastructure. Section 6 provides the simulation performed with the LEO constellations (such as Starlink and OneWeb). The corresponding simulation results are also included. Section 7 provides a performance analysis of LEO and GEO satellites. The LEO spectral efficiency analysis is also presented. Finally, the main conclusions are drawn in Section 8.

## 2. Review of Satellite Usage in Railway

Ref. [3] provides a review of the most important features and impacts introduced by the new Satellite communication (SatCom) technologies, also known as Non-Terrestrial Networks (NTN), according to 3GPP taxonomy. They can provide innovative onboard processing techniques, space data acquisition, collection, and post-analysis. In terms of enabled applications, these new SatCom constellations enable the network integration with terrestrial networks (such as 4G/5G/B5G) and innovative applications in the field of earth observation and communications in the transport sectors, such as rail, road, aeronautics, and maritime. Furthermore, the paper underlines that the NTN can improve the 5G network reliability, especially in the case of moving platforms such as cars, trains, and airplanes. The Internet of Things (IoT) networks can also benefit from the NTN. The methodology adopted for the review is based on five items: system and architecture, air interface, medium access, networking, and test and prototype. Satellite Network Automation, Resource Orchestration, Quantum Key Distribution (QKD) through Optical SatComs, and Machine Learning-based applications are only some key innovative features of NTN.

Ref. [4] defines a techno-economic model to be used for the sustainability assessment of LEO adoption. The LEO constellations provided by Starlink, OneWeb, and Amazon Kuiper are analyzed in terms of Free Space Path Loss, Carrier-to-Noise ratio, and Channel capacity depending on the number of satellites. Finally, the mean capacity per subscriber density is also calculated. The main conclusions underline the role of LEO satellites in communication technologies. The analysis results show that these new SatCom constellations are suitable in the case of a maximum of 0.1 users per km^2^ if they intend to offer a connectivity service comparable to the other broadband solutions.

Ref. [5] is focused on LEO usage in the rail sector, especially in the case of the High-Speed Railway (HSR). The terrestrial networks (such as 4G) can offer a connectivity bitrate up to 100 Mbps covering a cell where HSR is moving up to 350 km/h. The link drop phenomenon and the high number of requested handovers are the main challenges to be managed. LEO systems are a valid alternative to terrestrial communication technologies also thanks to the implementation of content delivery able to guarantee an acceptable delay in the order to 20–30 ms. The authors define an integrated terrestrial-satellite network (ITSN) as providing high data rates and connectivity without interruptions. ITSN can integrate different communication technologies (also known as bearers) thanks to the implementation of the Multipath TCP (MPTCP) protocol. The role of caches is essentially to combine the traffic data and to serve the end-users proactively.

In [6], the Intelligent Train Tracking System (ITTS) is analyzed in terms of uplink Medium Access Control (MAC) protocol and the corresponding probability of train detection enabled by the usage of LEO satellites. In particular, a Very High Throughput (VHT) Medium Access Control (MAC) protocol is analyzed and defined as a discrete-time Markov Chain. The analysis results show that the VHT MAC protocol provides a maximum throughput better than Slotted Aloha (SA) MAC protocol. It makes ITTS more reliable, robust, and resilient to LEO satellites.

Ref. [7] presents a review of the 3rd Generation Partnership Project (3GPP). It also underlines the role of LEO SatComs as innovative communication technology supporting the terrestrial 5G networks and beyond. All the satellite vendors need to be involved more and more in the activities of 3GPP and other standardization bodies on 5G technology, especially 5G New Radio (NR). In the case of Sixth Generation (6G) systems, the LEO satellites play a crucial role since they can offer ubiquitous global connectivity.

In [8] the role of LEO in future autonomous transportation systems is described in terms of signal latency, cost, and performance. The paper provides a review of the main methodologies to optimize the LEO communication systems, focusing on the space segment, ground segment, user segment, LEO Networks, impact on autonomous vehicles, and cost/coverage issues. The authors also consider the high-speed trains to be served by 5G and LEO systems, together with Machine Learning-based approaches. They underline the important role of LEO communication systems for communication, navigation, and sensing functionalities in the autonomous transport sector.

In [9], the Second Generation Digital Video Broadcasting via Satellite (DVB-S2) is used for datacast transmission for railway applications requiring data integrity and guaranteeing the QoS targets. The paper proposes unicast features to support the mobility of devices. The authors define an innovative architecture based on two link-layer protocols for data encapsulation: Multiple Protocol Encapsulation (MPE) and Generic Stream Encapsulation (GSE). The paper provides a comparison of them and shows how extending the GSE headers to support the LL-FEC identifier descriptors.

## 3. Railway Scenarios and Applications

### 3.1. Railway Scenarios

S2R Ju identified several railway scenarios [2]. The railway scenarios are essentially grouped into five categories: mainline, urban/metro, regional, freight, station/yard/depot.

The mainline line is the line used by different trains and often it connects towns, representing the main route. In the last years, it was enhanced and upgraded to “High-speed Rail”. In this special mainline, the trains are moving at a higher speed than the traditional rail traffic, thanks to dedicated rolling stock and tracks. In High-speed Rail the trains can reach a speed of 200 km/h, even 250 km/h in some cases. This depends on the specific conditions of the railway track. In the future, a train is expected to reach a speed of up to 500 km/h. In this case, the safety and reliability of railway lines are crucial aspects to be managed carefully. Voice and data services are the main applications.

Urban and metro lines are included in Rapid Transit Railways (RTR) operating in urban and suburban areas. Therefore, they are characterized by public transport with high capacity. RTR lines have also the exclusive right of way and are allocated to specific assets, such as separated tunnels or elevated railways.

Regional railway lines operate out of the urban areas and provide transport services to the passenger connecting different towns and cities. There are several stops across the line.

Freight rail lines are characterized by low density and higher distances across several regions. They are often formed by single tracks and they are affected by the lack of modern communication technologies. They are still based on old safety systems and manual operations.

Finally, the station, yard, and depot are the last railway scenario categories. The station is a railway location where a passenger train can start, stop or end its journey. The yard is constituted of a group of tracks to arrange the trains (such as train-shunting procedure), and other purposes. The train depot is the location where trains are recovered and maintained.

### 3.2. Railway Applications and Traffic Classes

The ACS Traffic Class IDs have been defined in accordance with the FRMCS requirements [2,10]. Table 1 shows the ACS Traffic Classes, while Table 2 the FRMCS requirements.

The first priority is to avoid train derailment [11] and then to guarantee the security and safety of the train transport system [12]. Signaling application to support transport operation. Signaling information is exchanged between the train and the Control and Command Center as a burst-based message. The packets report the train position (Position Report, PR) and the corresponding authorization to move across the line (Movement Authority, MA). The complete mechanism is specified by the European Rail Traffic Management System/European Train Control System (ERTMS/ETCS). Critical voice application is performed by the train’s personnel to communicate with the other rail nodes and emergency entities. Multi-call can be also performed. Critical video applications are used for surveillance purposes. Critical data applications regard the exchange of data on the safety of the train and the considered rail line, such as the communication with the track-side or eventual railway interruption. Non-critical data applications regard the information on train or railway line maintenance and Passenger Information System.

## 4. ACS and FRMCS System Architecture

In the post-GSM-R era, two integrated approaches are defined to overcome the dismission of GSM-R systems. Both of them envisage the possibility of using different carriers as train-to-ground communication technologies to provide multiple connectivities.

The technical specifications of ACS are defined in [1,2]. The specifications define an overall system architecture able to use different communication technologies independently. This makes ACS a cost-effective and scalable communication system for managing multiple access networks and enabling the bearer’s independence principle.

The ACS also enables (i) efficient QoS control and management, (ii) throughput enhancement through carrier aggregation, and (iii) resilience improvement through data redundancy.

As shown in Figure 1, the ACS Gateway (ACS GW) is deployed both on the client-side (ACS Onboard GW, ACS OGW) and server-side (ACS network GW, ACS NGW).

From the bearer point of view, in ACS the Control plane and User plane are separated and the ACS GW interfaces with radio communication bearers at the IP level only. Moreover, the ACS GW performs the following actions: (i) setup of Tunnels over the IP bearers, (ii) intelligent control of bearers, and tunnel monitoring. As mentioned, the ACS creates the tunnels and manages the registration, the management of communication sessions, and the control of information and events services for the applications. From the application side, the single rail application can interface with the ACS control side and/or ACS user plane, as shown in Figure 1.

The ACS is one of the two communication systems for the railway sector. The other one is represented by the Future Railway Mobile Communication System (FRMCS) framework. In 2015, the ETSI Technical Committee for Rail Telecommunications (TC RT) started to work on the Next Generation Radio for Rail and to study the Future Railway Mobile Communication System. The main FRMCS technical standard documents are constituted by ETSI TS 103 764 and ETSI TS 103 765, where FRMCS System Architecture and corresponding building blocks and functions are specified, respectively, as shown in Figure 2.

The FRMCS client is built in the ACS directly in terms of ACS on the server network side and the corresponding app. Theoretically, the ACS should not interact with the MNO network, instead, it is an overlay infrastructure on top of the MNO network. The FRMCS, on the other hand, presupposes the direct interaction of the MNO because the FRMCS server interacts with the MNO network according to the service requirements requested by the terminals on board the train. Recently, it seems that ACS also foresees the possibility of having servers (as in the case of FRMCS servers) that interact with the MNO network.

Due to this approach, the two systems, ACS and FRMCS, will be identical from an architectural point of view. ACS uses IETF protocols also for signaling (SIP protocol), precisely because it performs an overlay transmission.

## 5. Framework for Performance Analysis

### 5.1. LEO Satellites and ACS

In the simulations, the LEO satellite constellations from Starlink and OneWeb companies are considered.

LEO systems are defined as satellites having elliptical/circular orbits ranging from 500 to 2000 km above the Earth’s surface and below the Inner Van Allen Belt. The orbit period can vary from 90 min to a couple of hours. The LEO system’s radius is from 3000 to 4000 km [13].

The launch of the first two Starlink prototypes of LEO satellites started in February 2018. In September 2022, Starlink launched nearly 3000 satellites. The second generation of Starlink LEO satellites will enable them to be connected to the smartphone directly [14].

Regarding OneWeb LEO systems, the constellation of 648 satellites will be completed by the end of 2022 [15].

### 5.2. Adopted Methodology

The scope of the performance analysis is to study and analyze the LEO signal in terms of SINR and related bit rates offered to the train, considering opportune modulation and code scheme (MCS). The adopted methodology is shown in Figure 3.

The workflow in Figure 3 is based on the necessity to compose the LEO Satellite scenario. It consists of:System modeling;Radio propagation model;Identification of the railway line;LEO Ground station coordinates;Starlink LEO satellites coordinate;Composition of the LEO satellite scenario;Simulations of the LEO constellation for the evaluation of the radio link budget.

The first step is to create the LEO satellite scenario to be considered for the simulation and analysis. To compose this scenario system, modeling is necessary to reference architecture, satellite terminal, earth ground station, and railway application. Moreover, the railway line allows a definition of a realistic scenario, as well as the opportune radio propagation model to be used for the radio link budget. The LEO satellite scenario is completed thanks to the possibility to import the coordinates of real existing LEO constellations (such as Starlink Starlink and One Web). The simulations are then performed to calculate the radio link budget and the corresponding SINR. The simulation results provide the SINR trend values of LEOs signals along the considered railway line.

In the present paper, the performance analysis is carried out at the system level.

In Figure 4 the considered system architectures for non-terrestrial networks (NTN) networks are shown [15].

Figure 4a depicts the reference architecture of solutions for New Radio (NR) to support non-terrestrial networks (NTN) with a transparent payload.

NTN system architecture with the transparent payload is usually constituted by the following elements:A satellite with a transparent payload and several beams. The Field Of View (FOV) of the satellite is depending on the onboard antenna diagram and minimum elevation angle.A transparent payload for filtering, converting, and amplifying the radio frequency, modulation, demodulation, coding, and decoding.The User Equipment (UE) is able to receive the satellite signal in the considered service area.Satellite gateways for the interconnection with a public data network.The Feeder link between the satellite gateway and the satellite.The service link between the User Equipment (UE) and the satellite (or UAS platform).

Figure 4b shows the considered LEO system architecture with the integration with ACS devices in the rail sector.

The LEO terminal and the satellite antenna are installed into the On Board Unit (OBU). They are both present in the ACS OGW. The Rail App is connected to ACS OGW through a Local Area Network (LAN) connection supported by Ethernet transmission protocol.

ACS OGW is connected to the LEO satellite in orbit through satellite radio access technology depending on the available modulation and Coding scheme (MCS).

As evidenced in Figure 4b, the LEO satellite can be connected to the Radio Block Center (RBC) according to three different possible schemes: (1) the RBC is connected to the 5G network and external data network (i.e., Wide Area Network (WAN) context), (2) the RBC is connected to the external data network (i.e., Wide Area Network (WAN) context), and (3) the RBC is connected directly to the ground station (gateway). The RBC is equipped with a network-side ACS NGW, capable of closing the tunnel (stable between ACS OGW and AVS NGW) and then transmitting the information flow to the server-side Rail App.

As regards the role of satellite nodes on the network, not having further details on the Starlink/OneWeb LEO transmission format, it has been hypothesized that the satellite is transparent and non-regenerative. That is, it is capable of transmitting to the ground station the information flow it receives from the ACS OGW on board the train.

From the point of view of access or switching protocols, if the ACS OGW is connected to the 5G network through a direct connection with the gNB (case (1) of Figure 4b), the service link realizes the Uu Access Interface. In the other cases ((cases (2) and (3)) of Figure 4b), the interconnections are based on the IP protocol.

Since the Modulation and Coding Scheme (MCS) is not known in the case of Starlink and OneWeb constellation, the DVB-S2+RCS protocol was considered, as specified in [16]. The performance evaluations concerned only the radio access network between the satellite and the OBU not in terms of success rate, as in the case of the 5G network [17].

Since the MCS for Starlink and OneWeb is not available, the performance analysis is based on the C/N + I and the corresponding data rate.

The main assumptions regard:The upstream and downstream paths are analyzed separately;The LEO operator fully manages the satellite-to-ground gateway and guarantees the necessary reliability and system capacity;The WAN connection between the ground station and the RBC is not analyzed.

The radio propagation model of the Digital Video Broadcasting DVB-S2 satellite communication system is used to perform the simulation, as in [18]. The analysis also takes into account the attenuation models defined in the recommendation ITU-R P.618-13 [19]. The Rome–Florence Italian railway line is considered to perform the analysis. It can accommodate the mainline, regional, and freight railway lines. The LEO Ground Stations are placed along the railway line connecting the cities of Rome and Florence. All the ground stations or Reference Points (RPs) are referred to through the corresponding geodetic coordinates. RBC server is located in another Italian city, Bologna. In Table 3 the geodetic coordinates of some ground stations are listed.

The Starlink and One Web LEO satellite geodetic coordinates are imported from the corresponding available Two Line Element (TLE) files [20]. Then, the LEO satellite scenario can be composed in terms of LEO satellites, ground stations, and radio propagation model. The simulations are mainly focused on the radio link budget calculations. In this sense, the main satellite parameters are to be calculated.

Starting from the azimuth, elevation, and range values of each satellite concerning each one of the selected ground stations or Reference Points (RPs), the following parameters are calculated:Best and worst case for LEO satellite in terms of visibility.Distance range of the best satellite versus time duration (during 24 h).Propagation delay of the best satellite. The best satellite is the visible satellite with a minimum distance from the reference point for each time interval.Propagation delay of the worst satellite. The worst satellite is the visible satellite with a maximum distance from the reference point for each time interval.Distribution of visibility intervals of all satellites for each RP. The satellite is visible when its elevation angle is greater or equal to 15°.Distribution of the visibility intervals of the best satellite for each RP.Link budget to calculate the C/(N + I) ratio.

The procedure used to evaluate the link budget for each RP considers the best, the worst, and the “persistent” satellites. The persistent satellite is defined as the satellite having the highest visibility time interval from the considered RP. When a specific satellite becomes no longer visible, the modem selects another satellite with its visibility time interval. The link budget is calculated also considering all the time intervals during the day. The procedure is detailed in the following points.

Calculations have been carried out considering the real geodetic coordinates of the Starlink/OneWeb LEO satellites and those of the Reference Points (RP).
Set the physical transmission/reception parameters of LEOs and RPs. As in [16], the transmission/ reception parameters used for the DVB-S2 system have been considered for calculations.Calculate the LEO-RP distances by simulating the movement of LEO satellites and assessing the satellites that are visible from each RP at the specified time instant.Given the LEO-RP distance, calculate the corresponding path loss (free space path loss model is considered).Calculate the corresponding C/(N + I). Receiver noise parameters and interference effects considered in the calculation are the same in [16].

The following Equation (1) describes how to calculate the radio link budget of the considered LEO satellite constellation [21,22]. The formula is in dB.
P_RX_ = P_TX_ + G_TX_ − L_TX_ − FSPL − L_M_ + G_RX_ − L_RX_(1)
where:P_RX_ [dBm]: received powerP_TX_ [dBm]: transmitter output powerG_TX_ [dBi]: transmitter antenna gainL_TX_ [dB]: transmitter lossFSPL [dB]: Free Space Path LossL_M_ [dB]: additional losses due to fading, body loss, atmospheric loss, etc.G_RX_ [dBi]: receiver antenna gainL_RX_ [dB]: receiver losses

The Carrier-to-(Noise + Interference) ratio is given by the following formula:SINR = C/(N + I)(2)
where N is the thermal noise power and I is the interference power received.

The Equation (2) allows calculating the following parameters:The ratio of the energy per transmitted symbol to single-sided noise power spectral density, given by the following Equation (3):E_s_/(N_0_ + I_0_)(3)
where E_s_ is the energy per useful symbol.The ratio between the energy per information bit and single-sided noise power spectral density is given by the following Equation (4):E_b_/(N_0_ + I_0_)(4)
where E_b_ = E_s_ − 10 × log10.

According to [23] the opportune modulation and coding scheme (MCS) are considered considering the target margin (in dB). The available bit rate is the parameter indicating the speed of the LEO communication link. It can be calculated from the Spectral efficiency (for example in terms of MCS) and the transmission bandwidth, as specified in the Equation (5):Available Bit Rate = Spectral Efficiency × bandwidth(5)

## 6. Simulation Results

The simulation results refer to the LEO system model defined in Section 5.1. The railway line from Rome–Florence is considered in the simulations. The RPs are placed along the mentioned railway line.

Figure 5 shows the number of LEO satellites (represented by the asteriks) from Starlink and OneWeb that are visible in Italy during the day, i.e., with an elevation angle greater than 15°. As from Figure 5, the number of Starlink satellites is almost constant during the day, except for some peaks. The number of OneWeb satellites is more variable during the day. As mentioned, it depends on the different deployment strategies of the two companies.

As from Figure 5, in the Rome–Florence railway line around n. 30–40 of Starlink LEO satellites are visible, n. 8 in the case of OneWeb. The number of visible LEO satellites depends on the different strategies and launch plans adopted by the two companies.

Figure 6 shows the distance (in km) of a couple of LEO Satellites visible from an RP (placed in Rome) in a specific time interval.

As from Figure 6, the distance ranges from 1200 to 600 km and from 2600 to 2400 km in the case of Starlink and OneWeb, respectively.

Figure 7 shows the distance range of a single LEO satellite visible during the day from two different RPs, one in Rome and one in Bologna, in the case of Starlink and OneWeb constellations. 

As from Figure 7, due to the limited distance of about 400 km from Rome to Bologna-Florence in terms of LEO systems coverage, the two different RPs see the same satellite.

For example, the same Starlink satellite can be visible at 8 a.m. at 1500 km in Bologna and 1200 km in Rome. The same OneWeb satellite can be visible at 1900 km in Rome, and at 2150 km in Bologna during the same time interval (from 1:30 and 2:00 a.m.).

Figure 8 plots the distance range of the best and worst LEO visible satellites during the day for Starlink and in a limited time interval for OneWeb. For Starlink, the second-best available satellite is also included.

In Figure 8, the distance ranges of the best and worst satellites are indicated in black and red, respectively. In the case of Starlink, the distance of the second-best available satellite is indicated in blue. For Starlink, the best satellite has a minimum distance variable from 300 to 600 km, while the worst one is from 1300 to 1550 km (even 1600 km). The second “best” satellite ranges from 300 to 800 km. For OneWeb, the best satellite is from 500 to 1600 km, while the worst satellite is around 2200 km.

In Figure 9, the Cumulative Distribution Function (CDF) of the propagation delay in the case of the best and worst LEO satellite visible during the day. In Figure 10, the Complementary CDF (CCDF) of the visibility range of both the best and persistent satellite seen by two RPs (one in Rome and one in Bologna). Both plots are referred to as Starlink and One Web.

As from Figure 9, Starlink best and worst LEO satellite provides a propagation delay of 1.8–1.5 ms, and 2−5 ms, respectively, in 50% and 80% of cases. One Web best and worst LEO satellite provides a propagation delay of 4.2–5.6 ms, and 4.5–6 ms, respectively, in 50% and 80% of cases.

Figure 10 shows the visibility of the best and most persistent satellites seen from two different PRs, in the case of Starlink. The blue/black curves refer to the persistent satellite seen in Rome, while the green and red curves to the best satellite seen in Bologna. The visibility duration is around 1.5 and 4 minutes in 80% and 50% of cases, considering the persistent satellite. Considering the LEO satellite, the visibility duration decreases to values under 30 and 60 s in 80% and 50% of cases.

Considering OneWeb LEO constellations, the visibility duration of the persistent satellite is around 6.8 and 6.5 min in 80% and 50% of cases. It decreases to 1–2 min and 2–3 min in 80% and 50% of cases.

Figure 11 and Figure 12 show the Cumulative Distribution Function (CDF) of the achievable C/(N + I) in the case of best, worst, and persistent satellites. In Figure 11 the plot refers to the downlink path, while in Figure 12 to the uplink path. The CDF takes into account the data collected from all the RPs placed along the considered railway line.

In Figure 11 the downlink path is analyzed. Considering the Starlink constellation, in the case of the best satellite, the C/(N + I) value of the best satellite value is around 18 dB and 17.7 dB, respectively in 80% and 50% of cases. It reaches a value of 15.4 dB and 15.2 dB in 80% and 50% of cases in the case of the worst satellite. Finally, the persistent satellite provides a C/(N + I) value of around 17.7 dB and 16.7 dB in 80% and 50% of cases.

Considering the OneWeb constellation, the best satellite provides a C/(N + I) value of around 13.8 dB and 12.5 dB in 80% and 50% of cases, and 11.5 dB and 10.5 dB in 80% and 50% of cases if the worst satellite is analyzed. Finally, the persistent satellite provides a value of C/(N + I) around 12.5 dB and 12 dB in 80% and 50% of cases.

In Figure 12 the uplink path is analyzed. Considering the Starlink constellation, the C/(N + I) is around 18.5 dB and 18 dB in 80% and 50% of cases, if the best satellite is considered. The values decrease to 13.5 dB and 13.2 dB in 80% and 50% of cases if the worst satellite is analyzed. Finally, the analysis of the persistent satellite provides a value of C/(N + I) around 17.5 dB and 16 dB in 80% and 50% of cases.

Considering the OneWeb constellation, the C/(N + I) is around 12.5 dB and 12 dB in 80% and 50% of cases if the best satellite is analyzed. It is around 10 dB and 9.5 dB in 80% and 50% of cases if the worst satellite is considered. Finally, the persistent satellite provides a value of C/(N + I) around 12 dB and 11 dB in 80% and 50% of cases.

## 7. GEO/LEO Comparison and LEO Spectral Efficiency Analysis

The performance of LEO constellations can be also analyzed through a comparison with GEO constellations in terms of path loss [24]. Figure 13 gives a graphic representation of the path loss achievable using GEO and LEO constellations by Starlink (a) and OneWeb (b), in the function of elevation degrees. For both of these LEO constellations, the path loss is calculated considering the best and worst LEO satellites.

Figure 13 compares the path loss achievable by using GEO and LEO satellites.

The path loss obtained by HEO satellites is in yellow, by the worst LEO in red and the best LEO in blank. 

According to [24], the path loss achievable by a GEO satellite at 2 GHz frequency ranges from 189 to 192 dB for elevation angles ranging from 10 and 90 degrees. The line-of-sight conditions are assumed and the atmospheric losses are responsible for small deviations of the mentioned values of path loss.

The LEO constellations from Starlink can provide a path loss ranging from 175 to 180 dB in the case of the worst satellite. This is the situation of the LEO satellites with a higher distance from the considered terrestrial RP. In the best case, the path loss ranges from 164 dB to 173 dB. This is valid for the closest satellite to the considered terrestrial RP.

The OneWeb LEO constellations can provide a path loss from 178 to 185 dB in the case of the worst satellite. In the best case, it ranges from 167 dB to 180 dB.

From the service requirements point of view, the rail applications in the future situation (“tomorrow”) will require more bandwidth than the today situation (“today”), as shown in Table 4 [1,2].

In tomorrow’s situation, the video applications for the rail sector will be more present and will require ever more challenging performance. A number of eight on board the train are considered only for UL. It means the transmission analyzed is from the ACS OGW to the ACS NGW. The video format is Video Low Quality (LQ) at 500 kbps. A single camera for video critical application at 500 kbps is also considered for only tomorrow’s situation. In today’s situation, it is not considered. A total number of 12 trains within the Rome–Florence railway line is also estimated.

As from Table 4, the Video LQ is not implemented in today’s situation. In tomorrow’s situation, it will be a reality and it will require a bit rate of 14.4 Mbps only for the UL link. The same is for the critical video. It will be only implemented in tomorrow’s situation and it will require a bit rate of 7.2 Mbps.

The GEO and LEO satellites provide different performances in terms of bit rate and one-way delay. Table 5 summarizes the main features of Geo and LEO satellites in terms of one-way delay and single link Bit Rate in Downlink (DL) and Uplink (UL).

As in [25,26], GEO satellites (as Inmarsat) provide a worse performance in terms of one-way delay (250 ms) than the LEO satellites (10–15 ms). The LEO satellite can guarantee a single link bit rate of around 100 Mbps and 20 Mbps in DL and UL, respectively.

From Table 4 and Table 5, it emerges that the LEO satellites can provide a bit rate of 100 and 20 Mbps in DL and UL, respectively. These values are sufficient to support the bit rate required by both video LQ and critical video applications for the rail sector.

There are several benefits and opportunities deriving from the usage of LEO HTS satellites. In terms of capacity, at present, the LEO HTS satellites can guarantee a sufficient capacity to absorb tomorrow’s service traffic of a lift number of trains.

From the latency point of view, the GEO satellites are not compatible with the new services provided by the ACS classes that are characterized by low transmission latency, such as Automatic Train Operation (ATO) and Autonomous Train Control (ATC). They are the most latency-sensitive rail applications. The LEO HTS satellites can provide a latency of 5–6 ms compatible with the requirements of rail applications mentioned in Table 2. If the control center can be served by one or at most two satellites, LEO HTS satellites are able to satisfy the requirements of low latency services. They can also satisfy the ultra-low latency requirements thanks to a single LEO HTS satellite (one-hop scenario) that can be seen from two different reference points (RPs) (as in the case of the Rome-Florence rail section). It means in this case the single LEO HTS can serve both the rail control center and the ACS on board the train.

LEO HTS satellites can also improve the broadband internet connectivity destinated to the onboard rail personnel (Staff). The rail staff can use more bandwidth.

Finally, the trackside and wayside devices across the railway lines can also benefit from the usage of LEO HTS. They can improve the evolution of the trackside technologies used for monitoring the railway ecosystem, such as rolling stock, and tracks. They can also enable the interconnection with the trackside sensor network and they can contribute to the optimization of safety in specific points considered as critical, and, in general, the traffic management in the railway sector.

The performance of LEO constellations can be also analyzed in terms of channel utilization efficiency and then spectral efficiency.

The spectral efficiency η is given by the ratio in Equation (6):(6)$$\eta =\frac{C}{B}$$
where C is the Capacity, and B is the Channel bandwidth.

The relation between the spectral efficiency η and the SINR is given by the Shannon formula that in the more general case can be written as:(7)$$\eta =A\;\times \;\log_{2}\;(1+\frac{\mathrm{SIN}R}{G})$$
where G is the Gap and A is a constant that has been evaluated as follows.

As in [27], it has been observed that Starlink adopts OFDM modulation using modulations-coding-schemes (MCS) ranging from QPSK to 64 QAM and coding from 1/3 to 4/5.

Given the table of the minimum SINR required for each MCS and then the corresponding spectral efficiency, we have determined A and G values, as follows:(8)$$\overset{nMCS}{\underset{n=1}{\sum }}{\left[\eta_{\mathrm{MCS}\_n}-A\;\times \;\log_{2}\;(1+\frac{\mathrm{SIN}R\_n}{G})\right]}^{2}$$
where η_MCS_n_ is the spectral efficiency achieved when the n-th MCS is considered and the SINR_n is the corresponding minimum required SINR.

The MCS data used in (8) are reported in Table 6 and we have obtained A = 0.8368 and G = 1.4.

In Figure 14 and Figure 15 we plot the CDF of the spectral efficiency evaluated in accordance with (7) where A and G are given by solving the minimization of (8).

The best, persistent, and worst satellites both in downlink (DL) and uplink (UL) have been considered. The plot has been obtained using (8) and considering the SINR depicted in Figure 11 and Figure 12. 

In Figure 14a we plot the CDF of spectral efficiency in (7) for the downlink of the best, persistent and worst Starlink LEO satellites are depicted.

As from Figure 14, In Starlink, the best satellite provides a spectral efficiency ranging from above 3.1 dB which corresponds to a minimum MCS using 16QAM modulation format with a code rate of 2/3. As expected, Oneweb provides a low spectral efficiency corresponding to the 16QAM with a code rate of 1/3 up to 1/2. These values will be improved with the increase the number of LEO satellites to be launched in the next years and the use of lower satellite orbits. 

In Figure 15 the CDF of the best, persistent and worst LEO satellites in the uplink are depicted. The behaviour of LEO satellites from Starlink is shown in Figure 15a, Oneweb LEO satellites in Figure 15b.

Considerations similar to those carried out for spectral efficiency in downlink can be applied in the uplink case.

## 8. Conclusions

The railway sector has been characterized by the introduction of different communication technologies able to control and manage trains. The train-to-ground communications have been based on the terrestrial communication GSM-R. Since it is expected to be dismissed by 2030, the ACS and FRMCS constitute the future communication technologies in the rail sector. Beyond the terrestrial 4G/5G technologies, new LEO satellite constellations play a crucial role.

The objective of the present work is to suggest that the use of LEO HTS satellites for railways can constitute valid alternative bearers for the new ACS/FRMCS communication systems used in the railway sector. These intelligently use all the communication technologies available in a certain place and at a certain time.

What we have demonstrated in this article is that the new LEO HTS satellites are able, by themselves (and even in extreme situations, where the satellite is the only technology available), to support traffic for today and, above all, for tomorrow’s scenarios. More challenging railway applications will be increasingly pervasive, including the new ATO/ATC railway automation services.

The paper analyzes the performance in terms of available C/(N + I) and the bit rate offered by LEO satellites. Starlink and OneWeb constellations have been analyzed. The minimum elevation angle is 15 degrees. The aggregated traffic capacity depends on the Modulation and Coding Scheme (MCS). It can range from 800 Mbps to some Gbps.

To calculate the Bit Rate, the DVB-S2 modulation schemes are considered. The product of spectral efficiency with the Bandwidth gives the available minimum bit rate.

The simulation results are performed considering the Rome–Florence railway line. They evidence the presence of a group of LEO satellites covering Italy and the railway line. The number of LEO satellites depends on the LEO operator strategy, from 30–40 satellites by Starlink to 8 by OneWeb. The minimum distance ranges from 500 km to 1200 for Starlink and OneWeb satellites, respectively.

Due to the distance of 400 km between Rome and Bologna (where the RBC is located), the PRs along this railway see the same satellite.

In terms of propagation delay, it can vary from 1.8–1.5 ms, and 2–5 ms, respectively, in 50% and 80% of cases considering Starlink’s best and worst LEO satellites. Considering One Web’s best and worst LEO satellites, it can vary from 4.2–5.6 ms, and 4.5–6 ms, respectively, in 50% and 80% of cases.

In terms of C/(N + I) value, Starlink provides downlink/uplink values from 18/18.5 dB (best satellite), 15.4/13.5 dB (worst satellite), and 17.7/17.5 dB (persistent satellite).

One Web provides downlink/uplink values from 13.8/12.5 dB (best satellite), 11.5/10 dB (worst satellite), and 12.5/12 dB (persistent satellite).

Providing a considerable capacity enables LEO systems to manage a high quantity of data. This is very important in the case of bandwidth-demanding specific ACS traffic classes. Moreover, this makes LEO technology a competitor of terrestrial communication technologies, such as 4G/5G, where they are not available in a specific area.

Moreover, the analysis shows that the LEO satellite connectivity service is always guaranteed to the considered RPs. However, the LEO constellations are affected by the typical issues related to the satellite systems, such as geomorphology, tunnels, or other physical obstacles. These issues have to be considered to model an efficient LEO satellite service.

The high data rate and the large radio coverage offered by the LEO technologies can also guarantee high levels of connection reliability. This feature is determinant in the case of mission-critical applications. The LEO satellite operator is committed to managing the QoS/QoE aspects of the services. Future analysis needs to take into account the second generation of LEO satellites promising high performance in terms of coverage and data rate offered to the end user. The handover mechanism and the service continuity have to be evaluated according to the next generation of LEO satellites.

## Figures and Tables

**Figure 1 sensors-23-00068-f001:**
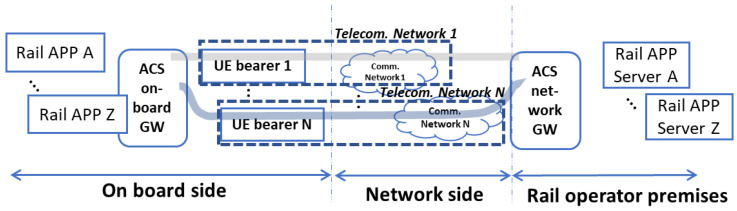
Adaptable Communication System (ACS) architecture.

**Figure 2 sensors-23-00068-f002:**
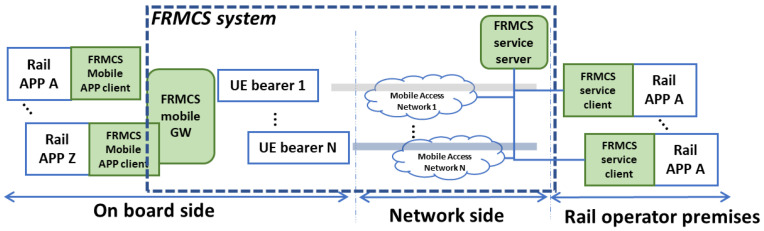
Future Railway Mobile Communication System (FRMCS) system architecture.

**Figure 3 sensors-23-00068-f003:**
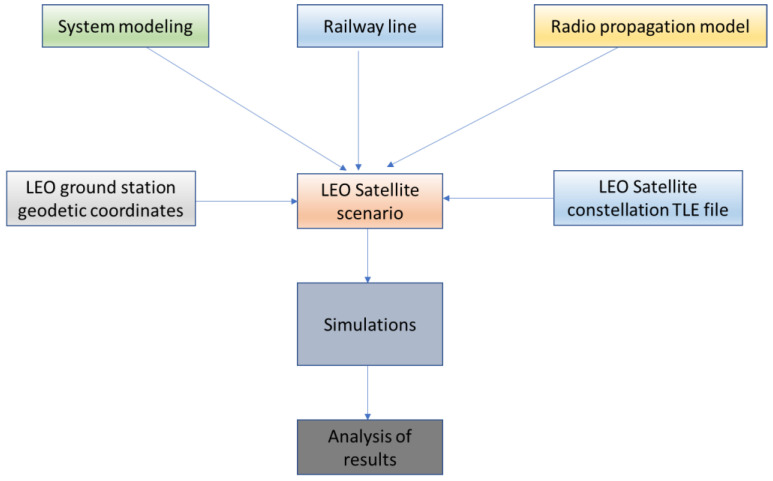
The adopted methodology.

**Figure 4 sensors-23-00068-f004:**
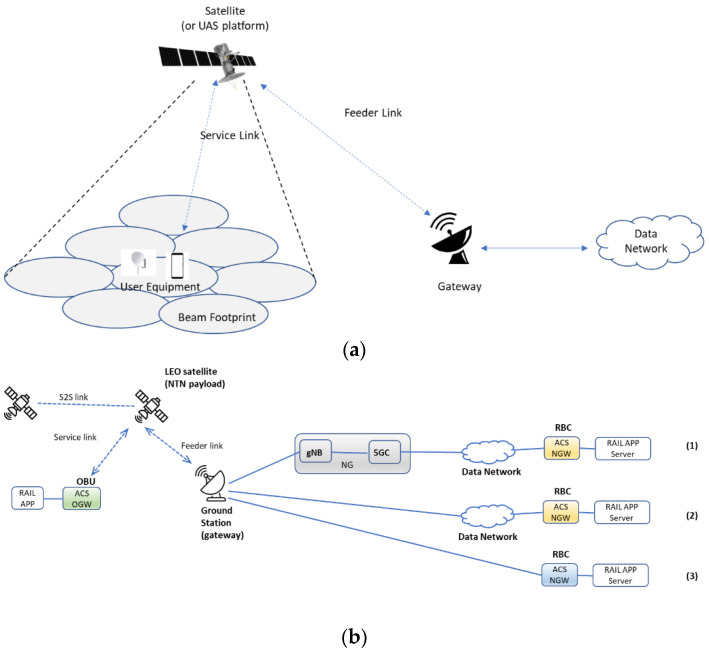
Non-terrestrial networks (NTN) with transparent payload. 3GPP-based solutions for NR (**a**), The considered LEO system architecture (**b**).

**Figure 5 sensors-23-00068-f005:**
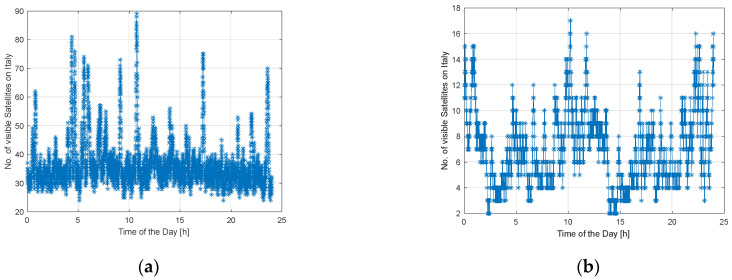
The number of LEO Satellites visible during the day: Starlink (**a**), OneWeb (**b**).

**Figure 6 sensors-23-00068-f006:**
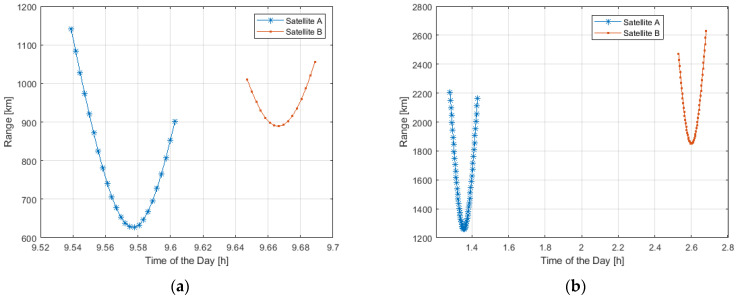
Range distance of a couple of LEO Satellites visible during the day from RP located in Rome: Starlink (**a**), OneWeb (**b**).

**Figure 7 sensors-23-00068-f007:**
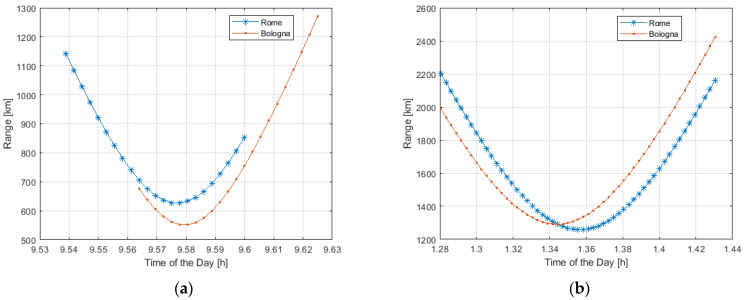
Range distance of the same LEO Satellite visible during the day from two Ground Stations (Rome and Bologna): Starlink (**a**), and OneWeb (**b**).

**Figure 8 sensors-23-00068-f008:**
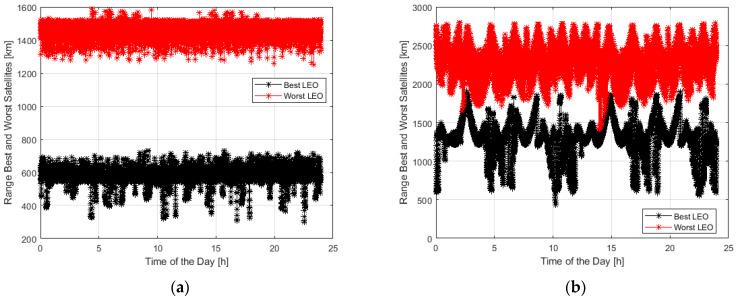
Range distance of the best and worst LEO Satellite visible during the day: Starlink (**a**), OneWeb (**b**).

**Figure 9 sensors-23-00068-f009:**
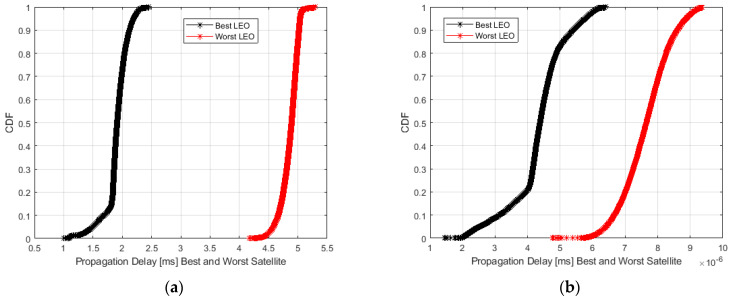
Cumulative Density Function (CDF) of the propagation delay in the case of the best and worst LEO satellite visible during the day: Starlink (**a**), OneWeb (**b**).

**Figure 10 sensors-23-00068-f010:**
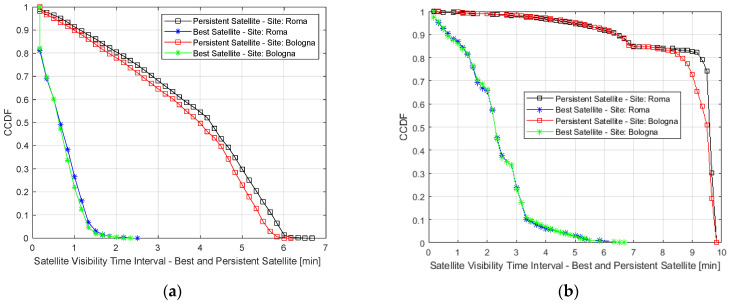
Complementary Cumulative Distribution Function (CCDF) of the visibility range of the best and most persistent satellite in visibility, seen from two different Ground Stations (Rome and Bologna): Starlink (**a**), OneWeb (**b**).

**Figure 11 sensors-23-00068-f011:**
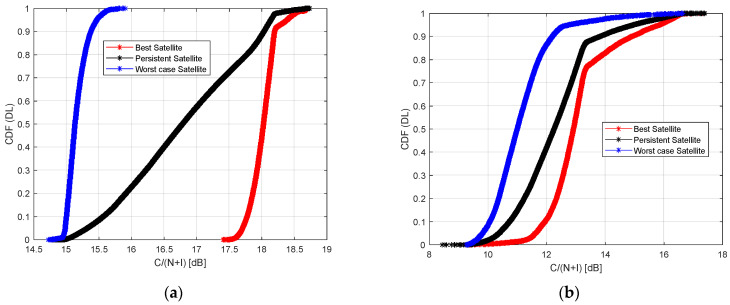
Cumulative Distribution Function (CDF) of the C/(N + I) in case of best, worst, and persistent satellites, visible by all Ground Stations, in downlink: Starlink (**a**), OneWeb (**b**).

**Figure 12 sensors-23-00068-f012:**
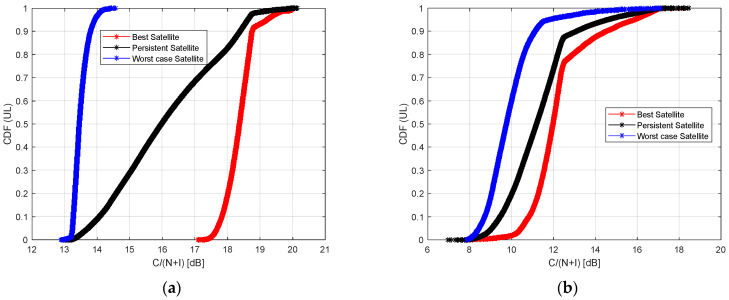
Cumulative Distribution Function (CDF) of the C/(N + I) in case of best, worst, and persistent satellites, visible by all Ground Stations, in uplink: Starlink (**a**), OneWeb (**b**).

**Figure 13 sensors-23-00068-f013:**
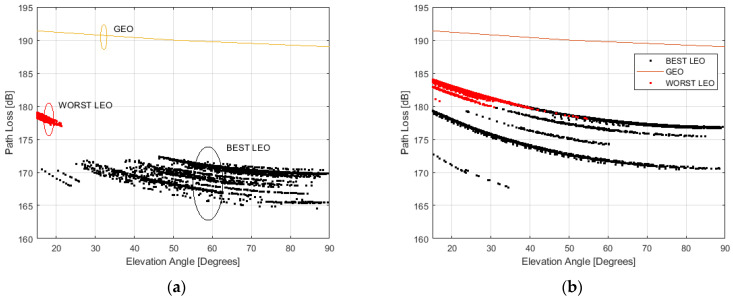
Comparison of GEO and LEO constellations in terms of path loss. Starlink (**a**), OneWeb (**b**).

**Figure 14 sensors-23-00068-f014:**
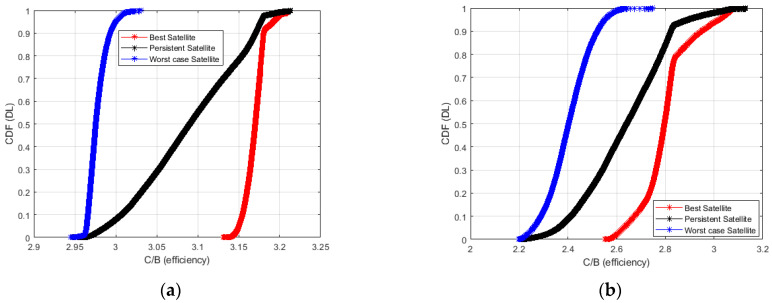
Cumulative Density Function (CDF) of spectral efficiency for the best, persistent and worst satellites in case of downlink transmissions: Starlink (**a**), OneWeb (**b**).

**Figure 15 sensors-23-00068-f015:**
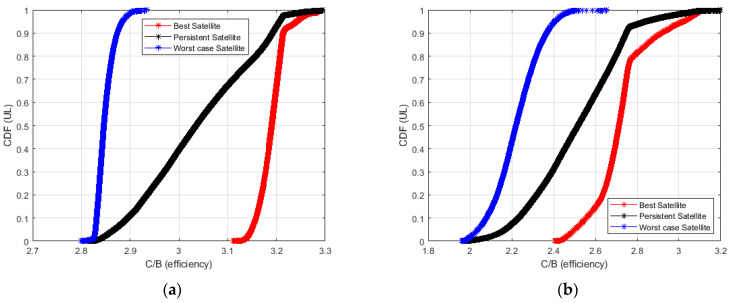
Cumulative Density Function (CDF) of spectral efficiency for the best, persistent and worst satellites in case of uplink transmissions: Starlink (**a**), OneWeb (**b**).

**Table 1 sensors-23-00068-t001:** ACS Traffic Classes.

Traffic Class ID	Reference FRMCS Application Category	Latency	Reliability	Setup Time
0	ACS control plane	FFS (for further study)	FFS	FFS
1	Voice	Low	Normal	Normal
2	Critical Voice/Critical Video	Low	Low	Immediate
3	Video/critical data (legacy apps)/Non-critical data	Normal	Low	Normal
4	Very Critical data	Ultra-Low	Ultra-Low	Immediate
5	Critical data	Low	Ultra-Low	Immediate
6	Messaging	Best Effort	Low	Normal
7	File transfer	Best Effort	Normal	Normal

**Table 2 sensors-23-00068-t002:** FRMCS Service Requirements.

Link Quality	FRMCS–Functional Requirement	FRMCS–System Requirement	Service Attribute Value	Impact
Latency	Low	Ultra-Low	≤10 ms	Impact on service responsiveness (i.e., mission-critical services)
Latency	Low	Low	≤100 ms	
Latency	Normal	Normal	≤500 ms	
Latency	Normal	Best Effort	>500 ms	
Packet Loss (%)	High	Ultra-Low	1–99.9999%	Impact on service throughput
Packet Loss (%)	High	Low	1–99.9%	
Packet Loss (%)	Normal	Normal	1–99%	

**Table 3 sensors-23-00068-t003:** Geodetic coordinates of some ground stations along the Rome–Florence railway line.

Station	Latitude–Longitude [Degree]
Roma, Termini	41.90134, 12.50034
Arezzo	43.47592731434826, 11.797549853039794
Firenze, Santa Maria Novella	43.76737, 11.27318

**Table 4 sensors-23-00068-t004:** Rail application requirements.

Rail Application	Number of Trains	Today Bit Rate DL [Mbps]	Today Bit Rate UL [Mbps]	Tomorrow Bit Rate DL [Mbps]	Tomorrow Bit Rate UL [Mbps]
Video LQ @ 500 kbps	12	0.00	0.00	0.00	14.4
Critical video @ 500 kbps	12	0.00	0.00	0.00	7.2

**Table 5 sensors-23-00068-t005:** GEO and LEO comparison in terms of one-way delay and single link Bit Rate in Downlink (DL) and Uplink (UL).

Satellite	One-Way Delay [ms]	Single Link Bit Rate DL [Mbps]	Single Link Bit Rate UL [Mbps]
GEO	250	1; 4	1; 4
LEO	10; 15	100	20

**Table 6 sensors-23-00068-t006:** Different MCS values used for LEO performance analysis.

MCS	η_MCS_	Minimum SNR [dB] per MCS
QPSK	0.67	1.00
QPSK	1.00	2.00
QPSK	1.33	4.30
QPSK	1.50	5.50
QPSK	1.60	6.20
16QAM	2.00	7.90
16QAM	2.67	11.30
16QAM	3.00	12.20
16QAM	3.20	12.80
64QAM	4.00	15.30
64QAM	4.50	17.50
64QAM	4.80	18.60

## Data Availability

Not Applicable.

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
