# Peer review of "Future Technologies for Train Communication: The Role of LEO HTS Satellites in the Adaptable Communication System"

_sensors, 2022, doi:10.3390/s23010068_

Round 1

Reviewer 1 Report

This paper presents the application of low orbit HTS satellite in the adaptive communication system of train. The authors described the communication needs of future trains and the role that LEO satellites will play, as well as the link budget. This is a very important application scenario for low orbit satellites. The techniques mentioned in this article were relatively important. At the same time, the expression of the article is clear. However, further optimization and explanation are needed in many aspects. The article cannot be considered for publication in its present form. The modification suggestions are as follows:

1) It needs to be made clear that the new generation of low-orbit satellites are mainly mobile communications satellites, such as Oneweb and Starlink, rather than broadcast satellites such as DVB. Therefore, when considering the system design, the analysis should be based on the 5G NTN benchmark.

2) When integrating low-orbit satellites into train ACS, the design needs to be more detailed. Further consideration discusses the network architecture, such as the role of satellite nodes on the network, such as whether satellites are fused to the ACS according to access or switching protocols. Further design is currently lacking.

3) In the simulation, only the basic constellation performance of the low-orbit satellite is presented, such as visibility, signal-to-origin ratio and so on. The author needs to further show the performance of low-orbit satellite applied to train ACS and whether it is feasible, such as access success rate, switching efficiency and so on.

4) At present, we only see the data and results in the paper, which are presented by the formula calculation. It is not possible to see whether there is a specific simulation process, whether there is a specific measurement result.

5) At the same time, the author needs to show the performance improvement brought by the adoption of LEO satellites compared to the adoption of ground network or high orbit, or the original LEO communication system.

6) There are several quotation errors in the text, please correct them.

7) The matching degree between satellite and train is poor in relevant literature researches.

Author Response

  • It needs to be made clear that the new generation of low-orbit satellites are mainly mobile communications satellites, such as Oneweb and Starlink, rather than broadcast satellites such as DVB.

According to the paper [17], DVB-S2 has not been considered as the transmission waveform for Starlink/OneWeb. As in [17], we only considered its parameters for evaluating the radio link budget in Starlink/Oneweb LEO constellations.

Therefore, when considering the system design, the analysis should be based on the 5G NTN benchmark.

We agree with you, but to properly answer this question we clearly need to know what are the multi-access protocols used by Starlink/Oneweb to allow users to access their network. But these protocols have never been disclosed to the open audience. Only very limited facts are known about Starlink/Oneweb. As an example, in October 2022 one researcher made reverse engineering on the structure of the radio signals received by LEO satellites on the ground (downlink only) and he published the results. But nothing more is now available. They do not consider the Starlink or OneWeb satellites. For this purpose, we added the ref. [19] where general LEO satellites are analyzed considering the 5G transmission system.

Figure 4. We modified the system architecture according to the 3GPP 5G NTN reference architecture. We also added the related references.

  1. Oneweb and starlink are not specifically designed for mobile users (the terminals are fixed, they are currently extending and studying the possibility of integration with mobile terminals).

From commercial offers published by Starlink, it seems for the moment that the main target of Starlink connectivity is fixed users mounting the antenna on the rooftop or in the garden. What we know is that some people have mounted the antenna on their caravan to assess if the satellite connectivity can be tracked at slow or moderate speed. We believe that Starlink is designing a new generation of new modems able to track the satellites even under general mobile conditions.

  1. the radio interfaces of Oneweb and Starlink are not known because they are protected by industrial secrecy and many newspapers (including those mentioned in the literature) have always assumed that at least the radio connection parameters are those of DVB and not those of 5G NTN, whose standardization is still in progress, there are no satellites implementing this interface nor the technical specification.

2) When integrating low-orbit satellites into train ACS, the design needs to be more detailed. Further consideration discusses the network architecture, such as the role of satellite nodes on the network, such as whether satellites are fused to the ACS according to access or switching protocols. Further design is currently lacking.

Starlink/OneWeb have not been designed having in mind the integration the 5G terrestial netwrok.

As an example, the Starlink LEO satellites do not implement any kind of 5G g-NB (full or partial).

They are proprietary systems. So, at the moment, it is very difficult (if not impossible) to make a comparison of a 5G NTN. In the revised paper, in Figure 4 we suggest that Starlink/OneWeb LEO satellite could be integrated at earth gateway level.

We added the following  text

-------Text added----

LEO terminal and the satellite antenna are installed into the On Board Unit (OBU). They are both present in the ACS OGW. The Rail App is connected to ACS OGW through a Local Area Network (LAN) connection supported by Ethernet transmission protocol.

ACS OGW is connected to the LEO satellite in orbit through satellite radio access technology depending on the available modulation and Coding scheme (MCS).

The LEO satellite can be connected to the Radio Block Center (RBC) according to three different possible schemes: (a) the RBC is connected to the 5G network and external data network (i.e. Wide Area Network (WAN) context), (b) the RBC is connected to the external data network (i.e. Wide Area Network (WAN) context), and (c) the RBC is connected directly to the ground station (gateway). L’ RBC è equipaggiato con un ACS NGW lato net-work, in grado di chiudere il tunnel (stabilito tra ACS OGW e AVS NGW) e quindi di trasferire il flusso informativo al Rail App lato server.

As regards the role of satellite nodes on the network, not having further details on the Starlink/OneWeb LEO transmission format, it has been hypothesized that the satellite is transparent and non-regenerative. That is, it is capable of transmitting to the ground station the information flow it receives from the ACS OGW on board the train.

From the point of view of access or switching protocols, if the ACS OGW is connected to the 5G network through a direct connection with the gNB (case a) of Figure 4), the service link realizes the Uu Access Interface. In the other cases ((cases b) and c) of Figure 4), the interconnections are based on the IP protocol.

Since the Modulation and Coding Scheme (MCS) is not known, the DVB-S2+RCS protocol was considered, as specified in [16, 17]. The performance evaluations concerned only the radio access network between the satellite and the OBU and concerned both the C/N+I and the corresponding data rate.

-------Text added----

  • In the simulation, only the basic constellation performance of the low-orbit satellite is presented, such as visibility, signal-to-origin ratio and so on. The author needs to further show the performance of low-orbit satellite applied to train ACS and whether it is feasible, such as access success rate, switching efficiency and so on.

Thank you for the comment.

We added the ref [19] where they evaluated the access success rate for 5G NTN LEO satellites, considering a 4-steps and 2-steps procedure. But in order to assess the access success rate, they explicitly considered the protocols that have been proposed for the 5G systems.

In the paper in questio,n it is calculated for 5G, but in the case of satellit,e it is necessary to know the procedure for accessing the satellite network, which in the case of Starlink/Oneweb is not made public.

The Switchig efficiency also depends on the particular LEO system to be considered. As mentioned before, the managed procedures and functions for LEO constellations performed by Starlink/OneWeb are not known at the moment.

4) At present, we only see the data and results in the paper, which are presented by the formula calculation. It is not possible to see whether there is a specific simulation process, whether there is a specific measurement result.

We added the following text in Section 5.2

-------------added text-------------------

The procedure used to evaluate the link budget for each RP considers the best, the worst, and the “persistent” satellites. The persistent satellite is defined as the satellite having the highest visibility time interval from the considered RP. When a specific satellite becomes no longer visible, the modem selects another satellite with its own visibility time interval. The link budget is calculated also considering all the time intervals during the day. The procedure is detailed in the following points.

Calculations have been carried out considering the real geodetic coordinates of the Starlink/OneWeb LEO satellites and those of the Reference Points (RP).

  1. Set the physical transmission/reception parameters of LEOs and RPs. As in [18], the transmission/ reception parameters used for the DVB-S2 system have been considered for calculations
  2. Calculate the LEO-RP distances by simulating the movement of LEO satellites and assessing the satellites that are visible from each RP at the specified time instant;
  3. given the LEO-RP distance, calculate the corresponding path loss (free space path loss model is considered)
  4. Calculate the corresponding C/(N+I). Receiver noise parameters and interference effects considered in the calculation are the same in [18].

-------------added text-------------------

5) At the same time, the author needs to show the performance improvement brought by the adoption of LEO satellites compared to the adoption of ground network or high orbit, or the original LEO communication system.

In section 6 we have added the following elements:

  • a Figure (Figure 13) including graphs with the comparison of the LEOs and GEOs path loss
  • the corresponding comment
  • the reference [26] “5G from Space: An Overview of 3GPP Non-Terrestrial Networks”.

6) There are several quotation errors in the text, please correct them.

We corrected the quotation errors.

7) The matching degree between satellite and train is poor in relevant literature researches.

We added the following reference [7] “Datacast Transmission Architecture for DVB-S2 Systems in Railway Scenarioswhere the applicability of DVB-S2 systems to the GEO satellites is described in the rail sector. Leo satellites are not considered. 

Reviewer 2 Report

This paper introduced future communication technology of railway sector, and underline the important role of LEO satellites in this field. This paper used a simulation system containing Starlink SpaceX constellation, OneWeb constellation, and the Rome-Florence railway line, to evidence the LEO satellite’s performance. But this paper suffers for several serious limits, which makes it not probably publishable.

1. This article lacks innovation, and it is hard to get the author’s contribution to this research field.

2.The simulation lacks comparison, and the simulation results don’t show the advantage over the contemporary technology.

3.The simulation results lack in-depth analysis. Some text content is not consistent with the figure,for example in line 376.

4. Figures 6~9 are missing legends, where the meaning of the curves cannot be directly seen.

5.There are a few grammatical mistakes, for example Line 58,230 and so on.

Author Response

This paper introduced future communication technology of railway sector, and underline the important role of LEO satellites in this field. This paper used a simulation system containing Starlink SpaceX constellation, OneWeb constellation, and the Rome-Florence railway line, to evidence the LEO satellite’s performance. But this paper suffers for several serious limits, which makes it not probably publishable.

  1. This article lacks innovation, and it is hard to get the author’s contribution to this research field.

Innovation: use of LEO HTS constellations in the railway and integration in ACS (from a systemic point of view) and therefore also in FRMCS, which are the two new systems for the rail.

We agree with the reviewers on the need for an experimental campaign in the field on trains equipped with Starlink/Oneweb terminals (based on which of the two offers support for mobile communications), which we are trying to start in the next year 2023.

For now, we've limited ourselves to assessing power levels and coverage range across the board, because the protocol used by Starlink/Oneweb is not known (see responses to reviewer 1). And we have shown that the power levels are almost constant along the whole line and that the SNRs vary little during the movement of the train along the line (even changing satellites) (because the SNR excursions are very limited  this means that the quality of the LEO from Starlink/Oneweb satellite transmission for the section in question (very important line in Italy) remains practically unchanged along the line (always barring obstacles). The simulation lacks comparison, and the simulation results don’t show the advantage over contemporary technology.

The only comparison that could be made is with the GEOs (Inmarsat) already envisaged for ACS/FRMCS, which however offer performances that are not in line with the requirements of railway applications in the coming years, especially in terms of capacity and latency.

Inmarsat (satellite and narrowband system)

Satellite

One-way Latency [ms]

Throughput [Mbps]

LEO

10,15

5,15,25

GEO (Inmarsat)

250,300

1,4 (*)

(*) https://www.e-sat.fr/en/inmarsat

  1. The simulation results lack in-depth analysis. Some text content is not consistent with the figure,for example in line 376.

We added a detailed description of the simulation procedure. Sorry, we modified the sentences in the mentioned lines.

  1. Figures 6~9 are missing legends, where the meaning of the curves cannot be directly seen.

We modified the mentioned figure with the insertion of legends

  1. There are a few grammatical mistakes, for example, Line 58,230 and so on.

We corrected the grammatical mistakes.

Reviewer 3 Report

The authors investigate a very interesting and important issue of using future communication systems for connectivity with High Throughput Satellites (HTS) and Low Earth Orbit (LEO) in the rail sectors. The paper analyses the LEO constellations of Starlink SpaceX and OneWeb using public data.

Comments: 

I propose that authors should also consider the requirements on broadband of satellite channels for the achievement of higher bitrates e.g., 1Gb/s from the point of view of the channel utilization efficiency (bitrate versus channel bandwidth) in comparison of both satellite systems (Starlink vs OneWeb). 

They should also consider the influence of a higher number of connected customers (railways) on the achievable bit rates for one customer. 

I propose to give attention to the utilization of results also for broader geographical areas (geomorphology, location of railways, etc.) because railways need very safe a reliable technology for their unfailing and secure operation.  At least they should briefly discuss possible obstacles to the use of their results.

Is Table 2. "FRMCS requirements", written correctly? In my view, there is a missing column about “What do the parameters in the individual lines represent (correspond to what)?”.

Author Response

The authors investigate a very interesting and important issue of using future communication systems for connectivity with High Throughput Satellites (HTS) and Low Earth Orbit (LEO) in the rail sectors. The paper analyses the LEO constellations of Starlink SpaceX and OneWeb using public data.

Thanks for the comment that highlights the simulations are based only on publicly available data, which are few, given that the others (more important for refining the evaluations) are absolutely not available to the public and in the literature. In the paper [18] he makes a basic analysis using the DVB-S2 parameters to evaluate the quality of the connection of the Starlink/Oneweb satellites.

Comments: 

  • I propose that authors should also consider the requirements on broadband of satellite channels for the achievement of higher bitrates e.g., 1Gb/s from the point of view of the channel utilization efficiency (bitrate versus channel bandwidth) in comparison of both satellite systems (Starlink vs OneWeb). 

Thank you for the comment. We added a new section 7 with additional figures (Fig. 14 and 15).

We show the plot “spectral efficiency versus SINR” based on the MCS known in a magazine special article on Starlink LEO transmissions (we added Ref. 27). The DVB-S2 transmission system is considered since any other details on Starlink are known officially.

  • They should also consider the influence of a higher number of connected customers (railways) on the achievable bit rates for one customer. 

Thanks for the comment. See the previous answer. In our system, the customers are the trains and in particular the ACS GW on board each train. We can introduce this model because ACS can use multiple technologies and it can use satellite technologies, as an alternative to the case where terrestrial ones (e.g. 4G/5G) are not available in a given area.

  • I propose to give attention to the utilization of results also for broader geographical areas (geomorphology, location of railways, etc.) because railways need very safe a reliable technology for their unfailing and secure operation.  At least they should briefly discuss possible obstacles to the use of their results.

We have considered Italy and the route in question is very important, because it is very extensive, busy, and simultaneously covers several scenarios (stations, regional, mainline). To evaluate the problem posed by the reviewer, it is necessary to analyze a European corridor, and evaluate any obstacles. Anyway, we added some remarks in the conclusion section (sect. 8).

  • Is Table 2. "FRMCS requirements", written correctly? In my view, there is a missing column about “What do the parameters in the individual lines represent (correspond to what)?”.

In Table 2 we added a column containing the details on the impact of each FRMCS service requirement in terms of the Impact on service responsiveness (i.e. mission-critical services) or on service throughput.

Round 2

Reviewer 1 Report

The authors have made some modifications according to my suggestions last time. To some extent, the contents described in the article is more relevant to the actual technical problems. Here are some suggestions from the authors' revisions:

1) As for the next generation of mobile satellite system models, the authors keep mentioning that Starlink and Oneweb have not disclosed their models and parameters due to commercial issues. This problem needs to be considered from two aspects. On the one hand, the author's work does not involve the design of the PHY layer, and the parameters on the link have been mentioned in the public literature. On the other hand, for the design of satellite network, the scope of the author belongs to system-level simulation, in fact, 3GPP NTN38 series of standards have given the design index. Please refer it.

2) I don't quite understand that the author is still optimizing the satellite, or optimizing the train communication based on the satellite. In my opinion, the focus is still on satellite optimization. These simulations mentioned and their performance are only based on constellation coverage capabilities. There is no mention of the performance gains to train communication after the use of satellite. This situation deviates greatly from the topic and theme of the article. Here I suggest that the author may at least need to refer to the needs of future train communication from the previous mention, combined with his own proposal of how the satellite enhanced communication system can meet. For example, the authors obtain the relationship between visibility, delay, path loss, and the efficiency of train communication system.

Author Response

  • As for the next generation of mobile satellite system models, the authors keep mentioning that Starlink and Oneweb have not disclosed their models and parameters due to commercial issues. This problem needs to be considered from two aspects. On the one hand, the author's work does not involve the design of the PHY layer, and the parameters on the link have been mentioned in the public literature. On the other hand, for the design of satellite network, the scope of the author belongs to system-level simulation, in fact, 3GPP NTN38 series of standards have given the design index. Please refer it.

Thank you for the comment. We integrated figure 4 (a, b) (Section 5.2) according to the 3GPP TR 38.821 V16.1.0 (2021-05) and referred to the 3GPP NTN38 in the revised version of the manuscript. We underlined that the approach of our paper is at the system level.

 2) I don't quite understand that the author is still optimizing the satellite, or optimizing the train communication based on the satellite. In my opinion, the focus is still on satellite optimization. These simulations mentioned and their performance are only based on constellation coverage capabilities. There is no mention of the performance gains to train communication after the use of satellite. This situation deviates greatly from the topic and theme of the article. Here I suggest that the author may at least need to refer to the needs of future train communication from the previous mention, combined with his own proposal of how the satellite enhanced communication system can meet. For example, the authors obtain the relationship between visibility, delay, path loss, and the efficiency of train communication system.

Thanks for the comments. There is no mention of optimizing train communication. Indeed the satellite has been seen as an alternative technology and integrated through ACS with the others. In the paper, we want to discuss the opportunities offered by LEO satellites, in terms of latency and bit rate, compared to GEO/MEO, already used in the railway sector.

In section 7 we added Tables 5 and 6 where we put the Rail application requirements in terms of bit rate DL/UL, both for the today and tomorrow situations. We calculated the required bit rate for video applications in the future and we compared them with the GEO and LEO satellite’s features in terms of latency and bit rate.

It emerges that the actual GEO satellites are not suitable to support the required bit rate and delay. On the contrary, The LEO satellites can do it.

Reviewer 2 Report

1.The academic viewpoint of this paper needs to be more refined, because the viewpoint - the role of LEO in railway system - has been discussed before.

2.In introduction, author mentioned the limit the usage of GEO and MEO. It's necessary to cite references or data to support this view.

3.Some legends block the curves.

Author Response

  1. The academic viewpoint of this paper needs to be more refined, because the viewpoint - the role of LEO in railway system - has been discussed before.

We have expanded the details concerning the role of LEO in the railway system adding the tables 5 and 6 … (see del reviewer 1)

  1. In the introduction, author mentioned the limit the usage of GEO and MEO. It's necessary to cite references or data to support this view.

Thank you for the comments. We added some references [1,2] in the introduction section about the GEO/MEO satellite already analyzed in the rail context by the European railway community (Shift-to-rail, now EU Rail).

  1. Some legends block the curves.

We modified the legend position in Figures 14 and 15

Round 3

Reviewer 1 Report

I think the author has revised all my suggestions.